# Liquid Biopsy: The Challenges of a Revolutionary Approach in Oncology

**DOI:** 10.3390/ijms26115013

**Published:** 2025-05-23

**Authors:** Claudio Antonio Coppola, Simona De Summa, Giuseppina Matera, Brunella Pilato, Debora Traversa, Stefania Tommasi

**Affiliations:** Unità di Diagnostica Molecolare e Farmacogenetica, IRCCS Istituto Tumori “Giovanni Paolo II”, 70124 Bari, Italy; c.a.coppola@oncologico.bari.it (C.A.C.); s.desumma@oncologico.bari.it (S.D.S.); g.matera@oncologico.bari.it (G.M.); b.pilato@oncologico.bari.it (B.P.); d.traversa@oncologico.bari.it (D.T.)

**Keywords:** liquid biopsy, CTCs, cfDNA, ctDNA, cfRNA, tumor-educated platelets, precision medicine

## Abstract

Liquid biopsy has gained attention in oncology as a non-invasive diagnostic tool, offering valuable insights into tumor biology through the analysis of circulating nucleic acid (cfDNA and cfRNA), circulating tumor cells (CTCs), extracellular vesicles (EVs), and tumor-educated platelets (TEPs). In this review, we summarize the clinical use of liquid biopsies in cancer now and look forward to its future, with a particular emphasis on some the methods used to isolate the liquid biopsy analytes. This technique provides real-time information on tumor dynamics, treatment response, and disease progression, with the potential for early diagnosis and personalized treatment. Despite its advantages, liquid biopsy faces several challenges, particularly in detecting analytes in early-stage cancers and evaluating the tumor molecular fraction. Tumor burden, molecular fraction, and the presence of subclones can impact the sensitivity and specificity of the analysis. Recent advancements in artificial intelligence (AI) have enhanced the diagnostic accuracy of liquid biopsy by integrating data, and multimodal approaches that combine multiple biomarkers such as ctDNA, CTCs, EVs, and TEPs show promise in providing a more comprehensive view of tumor characteristics. Liquid biopsy has the potential to revolutionize cancer care by providing rapid, non-invasive, and cost-effective diagnostics, enabling timely interventions and personalized treatment strategies.

## 1. Introduction

According to Globocan Cancer Observatory, cancer is the second leading cause of death worldwide, with 407,240 new cases estimated in Italy in 2022 and 2,001,140 new cases estimated in the USA in 2024 [1,2]. The pathological and clinical heterogeneity of cancer underlines the necessity of developing a personalized treatment for patients. Before the advent of precision medicine, as mentioned by The Precision Medicine Initiative, most clinical treatments were configured for the “average patient” and were able to fit perfectly with some patients but were inappropriate for others. Conversely, precision medicine considers the different features of each individual from a genetic to an environmental point of view, personalizing the treatment in order to obtain a more successful therapy for each patient [3]. Specifically, the precision medicine concept was promoted after the birth of next-generation sequencing (NGS), which provided an avalanche of large-scale human genome databases [4]. Before the NGS era, the use of the Sanger method did not provide the amount of data and the sensitivity required to screen a gene set involved in a specific cancer type in everyday clinical practice [5]. Moreover, first-generation sequencing was only suitable for discovering a few mutation types and small rearrangements, such as substitutions and small insertions and deletions. On the other hand, NGS allowed for the simultaneous detection of a large amount of mutations, large genomic rearrangements, and even copy number variation at an incredible resolution, including in single samples with a low nucleic acid yield [6]. The greatest issue in the oncologic field regards metastasis, one of the eight cancer hallmarks presented by Hanahan and Weinberg in 2000 [7], and subsequently confirmed in 2011 [8] and in 2022 [9]. Liquid biopsy (LB) represents a non-invasive method that has emerged as a promising tool in oncological diagnostics. Unlike tissue biopsy, which is widely used in clinical practice, LB utilizes body fluids, mainly plasma, to detect cancer-related biomarkers. This technique confers several advantages in clinical practice, such as the capacity to monitor cancer dynamics over time, assess treatment response, and detect minimal residual disease or early recurrence [10]. LB is a minimal invasive procedure that uses different biological fluids, which makes it possible to identify specific tumor mutations and could be applied to different cancer stages; in contrast, tissue biopsy is an invasive procedure, consisting of the acquisition of a tumor tissue sample, which remains the gold standard for the identification of tumor mutations in clinical practice nowadays [11]. As is well known, tumors are complex and widely heterogeneous, and change from a spatial and temporal point of view. Taking these considerations into account, LB arises out of the need to track the tumor’s evolution in real time. Indeed, the most important advantage in LB usage is its capacity to be less invasive compared to tissue biopsy, which requires surgical intervention; this feature is fundamental even when the disease is impossible to operate because of, for instance, the tumor localization and/or the presence of different tumor sites [7]. The main analytes detected in LB are circulating tumor DNA (ctDNA), cell-free RNAs (cfRNAs), circulating tumor cells (CTCs), extracellular vesicles (EVs), and tumor-educated platelets, which have different properties with different possible diagnostics values.

The importance of the LB in oncology is growing thanks to the personalized treatment of advanced-stage cancer patients [12]. To date, tissue remains the gold standard material for biomolecular analysis, and LB sources (especially plasma) are only used as alternatives when tissue is not available or not analyzable [13]. As mentioned in the first issue of the official Journal of the International Society of Liquid Biopsy, today several clinical trials are underway to allow for the use of LB as the gold standard not only for its predictive potential but even for disease monitoring, early detection, and screening [14]. Specifically, LB approaches make it possible to detect analytes, which could contain genes and/or proteins with tumor-specific alterations and serve as a mirror of possible metastasis. Nowadays, several clinical studies are examining the prospect of identifying these biomarkers in all cancer stages, which may potentially allow for early diagnosis in the future [15]. Using the search terms “liquid biopsy” and “cancer” and filtering from 01/01/2020, clinicaltrials.gov displays 31 clinical trials regarding LB (Appendix A). By serially sampling biofluids, such as blood, plasma, urine, and saliva, clinicians could track the evolution of the tumor in real time, monitoring disease progression and providing insights into tumor heterogeneity and clonal evolution. This tool might have a crucial role in understanding the mechanism of drug resistance [16]. Moreover, LB is already integrated into clinical practice, depending on the kind of tumor, helping with the identification of actionable mutations and allowing for personalized treatment strategies. For instance, the presence of EGFR mutations in cfDNA represents a milestone for the introduction of the LB in clinical practice, guiding the use of tyrosine kinase inhibitors in non-small cell lung cancer (NSCLC) patients when tissue is not available [17,18,19]. Post-treatment surveillance using LB can also detect minimal residual disease (MRD), thereby predicting relapse before it becomes clinically apparent, allowing for timely intervention and improving patients’ outcomes [20].

Since LB is rapidly evolving as a non-invasive and dynamic tool in oncology and several reviews have explored its general principles and clinical applications, the novelty of this review lies in offering a comparative and updated overview of the main classes of analytes (CTCs, cfDNA, cfRNA, EVs, and TEPs) with an emphasis on the analytical and technological challenges of their isolation. Furthermore, we provide a critical perspective on recent innovations, including AI integration, multimodal strategies, and updating clinical guidelines.

## 2. Analytes in Liquid Biopsy

### 2.1. Circulating Tumor Cells (CTCs)

CTCs have been known since 1869, when Ashworth described them as some cells in patients’ blood similar to cells inside the primary tumor [21]. CTCs represent the main analytes detectable with LB and are certainly the most promising for entering clinical practice, especially thanks to technological progress [22]. CTCs are defined as cells that originate from the primary tumor (as single cells or in clusters) and acquire the ability to spread into the bloodstream or lymphatic system and reach and eventually colonize distant organs. These cells are becoming increasingly important as biomarkers in several types of cancer, especially in metastatic conditions [23]. The main limitations in the use of CTCs lie in their enrichment, purification, and isolation, due to the presence of blood cells and even because these cells are usually present in the circulation in proportion to the tumor volume, making their isolation at an early stage very limited [24]. Several platforms have been developed to detect CTCs, based on immune-affinity, biophysical properties, and microfluidic systems, but the only device approved by the FDA is the CellSearch^®^ for the analysis of blood samples collected from patients with metastatic breast cancer, prostate cancer, and colorectal cancer, in which the CTC enumeration has a prognostic value [25]. Specifically, CellSearch^®^ is an immune-affinity device consisting of two main processes: the first one is a sample centrifugation performed to eliminate blood components, while CTCs are detected using anti-EpCAM antibodies conjugated with magnetic ferrofluid beads; the second one is the immunofluorescence step, used to further purify CTCs from contaminant blood cells using anti-cytokeratin antibodies and DAPI nuclear staining. The cells are scanned to detect EpCAM+/Cytokeratin+/DAPI+/CD45− cells, which will be CTC candidates [26]. The limitation of this tool is that it can only individualize CTCs from epithelial, but not from mesenchymal tumors. Hence, it would be necessary to also identify mesenchymal markers, which are upregulated on CTCs after the Epithelial-Mesenchymal Transition (EMT), in order to solve this limit [27]. Equally, endothelial cells in the blood (and most likely other normal cell types as well) may have the same CTC markers, thus the detection of false-positive results may be another limitation of CellSearch^®^. Using immunomagnetic beads (IMBs) conditioned with graphene nanosheets (GNs), known as protein corona disguised immunomagnetic beads (PIMBs), Zhou and colleagues have created an immuno-affinity CTC isolation technology to enhance CTC enrichment. To prevent the absorption and subsequent detection of some non-specific proteins, these conditioned beads can be disguised with blood proteins. Through the interaction of biotin and streptavidin, PIMBs are conjugated with an antibody that targets CTCs. According to Zhou et al., PIMBs disguised with Human Serum Albumin (HSA) demonstrated a leukocyte depletion percentage of approximately 99.996%, obtaining 62 to 505 CTCs from 1.5 mL of blood from cancer patients [28]. There are also important physical properties that distinguish CTCs and blood cells. Indeed, CTCs are larger in size, have more mechanical plasticity, and possess different mobility properties compared to blood cells. These differences are exploited for CTC isolation and enrichment with “label-free” methods. While CTCs can be isolated from red blood cells (RBCs), plasma and platelets can be isolated using density centrifugation medium or lysis methods [29]. It remains challenging to separate CTCs and leukocytes, because it has been observed that, although most CTCs are larger than leukocytes, some CTCs could be smaller than white blood cells (WBCs) [30]. ScreenCell^®^ is a device used to isolate and sort cells by size from blood samples. This system includes a microporous membrane filter through which the blood flow passes, which allows for detection by the size of the tumor cells and can be set in order to detect CTCs [31]. It is important to note that while CTCs usually exist in singular cells in the blood, they can also be found aggregate in doublets and/or in clusters, which are more efficient metastasis initiators than single CTCs [32]. Thus, using a method that might break CTCs clusters in single cells could lead to the loss of cluster-specific information [33].

Whatever method is used for CTC detection and enrichment, an additional step is necessary to isolate pure CTCs and to perform molecular analysis. Hence, every platform needs to be coupled with single cell-sorting technologies, such as fluorescence-activated cell sorting or microfluidic platforms [34]. Moreover, single cell platforms could be fundamental for tumor heterogeneity studies. Di Trapani and colleagues have developed DEPArray™, an image-based cell-sorting technology, which is a device that combines microelectronics and microfluidics in automatic, in order to isolate a single cell from a heterogeneous sample. DEPArray™ is often coupled with CellSearch^®^; it is a single-use technology based on Dielectrophoresis, which uses a nonuniform electric field to trap and move cells different in size. Through this device, it is possible to target a specific cell type and isolate it from the whole cell population. The cells isolated can be displayed as single cells or in pools and can be collected in several support types, in order to perform the molecular characterization of the selected cells [35].

### 2.2. Cell-Free TNAs: cfDNA and cfRNA

cfDNA is a combination of nucleic acids released into the bloodstream by various processes, such as apoptosis, necrosis, and active secretion with extracellular vesicles, discovered and first described by Mandel and Mateis in 1948 [36]. cfDNA is found as double-stranded fragments of about 150–200 bp, although it has been detected as fragments from 50 bp (larger proportion) [37] up to 39.8 kb in patients with hepatocellular carcinoma (HCC) [38]. The cfDNA concentration in the blood of healthy adults is usually quite low, while in cancer patients it increases to up to 50 times the normal level. According to Mattox et al., taking into account the stage and variability of the disease, the concentration of cfDNA in patients with colorectal, pancreatic, lung, and ovarian cancer (stage I–III) is approximately 12.6–18.1 ng/µL, whereas the concentration in healthy people is approximately 1–10 ng/µL [39]. In the latter case, part of the cfDNA comes from tumor cells and is called circulating tumor DNA (ctDNA) [40]. Technologies based on digital PCR (dPCR) and next-generation sequencing (NGS) are pivotal in detecting mutations, copy number variations, and methylation patterns [41,42].

To date, there is no single ctDNA assay that is suitable for all specific clinical applications. One of the main limitations of ctDNA analysis relates to the pre-analytical and analytical parameters that can affect the accuracy and reproducibility of the results, which are influenced by the ctDNA fraction levels, which may depend on treatments such as chemotherapy, immunotherapy, and radiotherapy. These pre-analytical variables also depend on the different modalities of the assays, which are based on the type of alteration to be detected and the technology used [43]. Another issue regards the possibility of false-negative and false-positive results related to the amount of ctDNA in the patient’s plasma and to clonal hematopoiesis, respectively [44]. The outsized contribution of expanded clones of hematopoietic stem and progenitor cells (HSPCs) to blood cell production, called clonal hematopoiesis, increases with the age of the patients, who present somatic alterations in blood cells but are not affected by disease [45]. Despite these limitations, ctDNA is widely used in clinical practice for cancer genotyping.

A consistent number of studies have demonstrated the fundamental role of ctDNA. Currently, ctDNA itself is clinically used as a biomarker, especially in breast cancer [46,47,48,49,50,51,52,53,54]. In 2020, Bratman and colleagues showed how plasma ctDNA levels in triple-negative breast cancer patients change in response to pembrolizumab treatment, and how ctDNA levels may be a potential predictor of tumor progression of this kind of tumor [55].

The first application of ctDNA biomarker detection in clinical real life was in metastatic NSCLC, due to the need to identify resistance-associated mutations (such as T790M) and to detect sequence changes that could be targeted with third-generation TKIs. Later on, ctDNA has been used in advanced breast cancer to detect PIK3CA mutations as resistance to cyclin kinase inhibitors [56], and, more recently, for the detection of ESR1 mutations [57]. Indeed, ESR1 has been identified as a crucial biomarker of metastatic progression in patients with metastatic estrogen receptor-positive breast cancer, which gives a mechanism of resistance in patients treated with aromatase inhibitors [58]. ESR1 is essential in individuals with advanced or metastatic breast cancer, with an estimated prevalence of mutations of about 30–40%. Elacestrant, a selective estrogen receptor degrader (SERD), was shown to increase the PFS in patients with ESR1 mutations in a randomized phase III EMERALD study [57]. In January 2023, Elacestrant was approved by the FDA for the treatment of advanced breast cancer patients with ESR1 mutations developed after estrogen-deprivation therapy with aromatase inhibitors in combination with CDK4/6 inhibitor, also approved by EMA in July 2023 [57,59]. The 2023 ASCO [60] and 2020 ESMO recommendations update, which recently changed the ESCAT level for ESR1 from IB to IA [61], underline the importance of the analysis in LB over the tissue analysis for the first time. In conclusion, ESR1 represents the most recent LB application in oncology, and supports the crucial role in the study of new biomarkers in biofluids.

DNA methylation is a biological process with a crucial role as a marker in gene expression, chromatin organization, and cancer development and progression. Methylation patterns are widely different across cell types and even across individuals. Therefore, the ctDNA methylation status should reflect the methylation status of the tissue, providing information on the tumor characterization and progression [62]. Furthermore, ctDNA methylation could be used to classify the tumor in subtypes, offering the possibility of an even more personalized therapy [63]. Although third-generation sequencing (TGS) platforms, in particular Oxford Nanopore Technology (ONT), were first employed for long-read sequencing, they have lately become essential for studying methylation patterns, especially in cfDNA, which is known to include small fragments. ONT has recently been used for the sequencing of cfDNA long fragments in order to assess the presence of ctDNA in biofluid by looking at the methylation pattern [64].

The ESMO tumor-specific recommendations are summarized in Table 1 [43].

cfRNA was first discovered by Lo and colleagues in 1999 in patients with nasopharyngeal carcinoma [65] and circulating tumor RNA (ctRNA), and is now a promising biomarker in oncology as it can be detected in biofluids, such as blood, urine, and saliva [66]. cfRNA is found in the bloodstream as various RNA subpopulations [67] that are associated with lipids [68], proteins, or incorporated into EVs to avoid degradation [69]. Techniques including qPCR [70], ddPCR [71], and NGS are used to analyze ctRNA [72]. cfRNAs include different kinds of RNAs, such as microRNAs (miRNAs), transferRNAs (tRNAs), piwi-interacting RNAs (piRNAs), long non-coding RNAs (lncRNAs), and others, which originate from various tissues and cells [73]. cfRNAs are present in the bloodstream and other bodily fluids because they are attached to proteins or encapsulated in EVs, which prevent degradation [74]. Koi and colleagues compared serum samples from breast cancer patients to those from healthy patients, discovering that three circulating small RNAs were differentially expressed: tRF-Lys (TTT), miR-21-5p, and miR-23a-3p. These three small RNAs were all found to be upregulated in breast cancer patients, and two of them were found to be in high concentration in the EVs, indicating that small RNA may be a potent cancer biomarker [75]. Recently, Kim et al. examined the blood samples of 160 patients with colorectal cancer and found 187 RNAs that were differentially expressed and linked to the progression of adeno-carcinoma. They also showed how circulating RNA might be used to enhance early diagnosis and disease monitoring [76].

### 2.3. Extracellular Vesicles (EVs)

In the last decade, EVs have been observed as a new frontier of cancer LB. EVs are 30–5000 nm lipid-bilayer spheres, released from cells into the body fluids in both physiological and pathological conditions, and containing several types of molecules, such as proteins, nucleic acids, lipids, and metabolites. Over the years, EVs have been explored as the major mechanism for cell-cell and cell-environment interactions. EVs are divided into three subtypes, based on their biogenesis: exosomes, shed microvesicles, and apoptotic bodies [77,78].

It has been proved that some bioactive cargoes may facilitate tumorigenesis and tumor progression through the activation/deactivation of different processes, such as angiogenesis and immune suppression [79]. The main advantage in the isolation of EVs compared to CTCs and ctDNA is the quantity of biomolecules, which are present in large amounts in biofluids (~109 vesicles/mL) and are secreted by different types of living cells, providing even more important information about cell origin and tissue status compared to ctDNA, which only reflects information for apoptotic cells [80]. Another advantage to consider is the stability of EVs due to the lipid bilayer, which allows them to circulate stably under physiological conditions and in the tumor microenvironment, and is also useful for their detection, isolation, and storage [81]. However, a major challenge is the purification and isolation of EVs secreted by tumor cells, especially exosomes, since they represent only a small fraction of the total amount of EVs. Several methods have been developed to detect and isolate EVs, but their limited sensitivity, specificity, and low purity due to contaminants remain a challenge for clinical use [82]. EVs are isolated using techniques such ultracentrifugation [83], size-exclusion chromatography [84], and immunoaffinity capture [85].

Although the use of EVs in LB is emerging in the diagnostics field, there are no standardized methods for their isolation and characterization. Recently, Dhani and colleagues published a case report, in which they performed a test (called ExoVita^®^) in a 60-year-old healthy patient with acute pancreatitis. This test is based on the measurement of protein biomarkers in exosomes derived from cancer cells, and it allows, combined with germline mutations in KRAS and TP53, for the early diagnosis of pancreatic ductal carcinoma (PDAC) [86]. Less recently, Moon et al. selected developmental endothelial locus-1 protein (Del-1) as a candidate biomarker on the exosomes surface for the early detection of breast cancer. They identified patients with early-stage breast cancer, proving Del-1 as a promising biomarker able to distinguish breast cancer and benign diseases [87]. Jiang et al. evaluated miRNA derived from EVs into the plasma in order to distinguish small cell lung cancer (SCLC) and non-small cell lung cancer (NSCLC) at an early stage. The study showed how the two tumor types have a different miRNA profile, and specifically that miRNA-483-3p in EVs could have a potential diagnostic value for early-stage SCLC, while miRNA-152-3p and miRNA-1277-5p could be used for the diagnosis of early-stage NSCLC [88]. There is an increasing, promising use of EVs in clinical practice, particularly those present in urine. In 2018, McKiernan and colleagues demonstrated how ExoDx Prostate IntelliScore (EPI), a urine exosomes gene expression assay, could improve the identification of patients affected by prostate cancer, and reduce the number of unnecessary prostate biopsies performed. This test measures RNA levels of three genes (*PCA*, *ERG*, and *SPDEF*) in EVs isolated from urine samples with a 92% specificity and 34% sensitivity [89]. In 2019, an EPI test was approved by the FDA, representing a strategy to diagnose prostate cancer in LB [90]. In 2023, Serratì et al. examined the role of EVs as biomarkers for monitoring anti-PD1 response, as well as their involvement in cancer progression and immunosuppression in metastatic melanoma. They demonstrated that PD1-positive EVs derived from cancer tissues represent a promising tool for monitoring anti-PD1 response treatment and for detecting acquired resistance to therapy [91].

### 2.4. Tumor-Educated Platelets (TEPs)

Platelets are blood enucleate cells generated by megakaryocytes, hematopoietic cells in the bone marrow. Although platelets lack genomic DNA, they contain the whole spliceosome machinery and some RNAs, crucial for the splicing modulation upon appropriate stimulation from specific cells [92]. The role of clot cancer-mediated was first described by Jean-Baptiste Bouillaud in 1823 and confirmed by Trousseau in 1868 [93]. It is now known that the active surface of platelets promotes cross-talk with other cells, including cancer cells, which can “educate” the platelets, altering their RNA profile and making them an important biomarker in LB [94]. The mechanisms through which TEPs interact with CTCs or with other cells in the bloodstream are still unclear, but one of the most credited theories hypothesizes that they have the ability to surround the tumor cells, protecting them by shear stress and preventing immune system attack [95]. It has recently been observed that TEPs can enhance the progression and prognosis of patients affected by colorectal cancer (CRC), interacting with tumor-associated macrophages (TAMs) [96], and platelets have been shown to play a role in the induction of chemoresistance in ovarian cancer [97].

TEPs are also important biomarkers thanks to their RNA profile, which can be analyzed using the NGS approach (RNA-seq), and which could allow for the prediction of diagnosis, prognosis, and disease monitoring following treatment [98]. The main analytes isolated with LB approaches are summarized in Figure 1, and Table 2 summarizes the different isolation methods, highlighting their benefits, limitations, and cost-effectiveness.

### 2.5. Comparative Clinical Performance: Liquid vs. Tissue Biopsy

Beyond the analytical description of circulating biomarkers, a comparative evaluation with traditional tissue biopsy highlights the clinical potential and current limitations of LB approaches. Compared to traditional tissue biopsies, LB might capture genomic heterogeneity from multiple tumor sites simultaneously and allow for real-time monitoring of tumor evolution. For example, blood-based genotyping may detect mutations from both primary and metastatic lesions, potentially revealing targets that a single tissue sample might miss [58]. LB is also safer and faster, since a routine blood draw avoids the risks of an invasive biopsy and could deliver results in days rather than weeks. In particular, one study of advanced lung cancer reported an average turnaround of 9.6 days for plasma NGS vs. 36.4 days for tissue biopsy [98]. However, tissue biopsy remains the diagnostic gold standard for confirming malignancy and histology, and is still required when LB yields no information, since LB sensitivity drops with low tumor burden [58,99]. Despite these limitations (including occasional false negatives due to low ctDNA fractions and rare false positives from clonal hematopoiesis), LB has established itself as a crucial complementary tool in oncology, with growing clinical applications in non-small cell lung cancer (NSCLC), breast cancer, and colorectal cancer [58].

In NSCLC, LB has rapidly entered routine practice for molecular profiling. Plasma ctDNA testing for EGFR mutations in advanced NSCLC is an early success story: it enables targeted therapy (EGFR tyrosine kinase inhibitors) even when tissue samples are unavailable or insufficient. The concordance between plasma and tissue genotyping in NSCLC is high, especially for dominant driver mutations. For example, studies using droplet digital PCR report that plasma EGFR mutation detection in advanced NSCLC has a sensitivity of 76–82% and specificity of 88–100% compared to tissue analysis [99]. It has been observed that concordance rates improve with higher tumor DNA shedding: one trial noted that combining LB with tissue testing identified actionable mutations faster and in more patients, with 94–100% agreement on key mutations between the two methods [98]. Clinical guidelines now support LB: the International Association for the Study of Lung Cancer recommends plasma testing for EGFR and other markers at diagnosis or relapse, recognizing that it can expedite treatment decisions [99]. Indeed, using LB first can significantly reduce time to treatment and often circumvents repeat invasive biopsies for detecting resistance mutations (e.g., EGFR T790M) [98]. Cost-effectiveness analyses in NSCLC have further reinforced the value of LB. A Canadian study found that adding ctDNA testing to standard tissue profiling in stage IV NSCLC saved about $3065 (Canadian Dollars, CAD) per patient (by avoiding some costs of procedures and suboptimal therapies) [100]. Similarly, a German modeling study reported an incremental cost-effectiveness ratio of about €54,000 for integrating LB, and even identified scenarios (EGFR-mutant cases) where LB-guided care was cost-saving (dominant strategy) [101]. Overall in NSCLC, LB provides a rapid, sensitive means to guide precision therapy in real time, though a negative LB result must be followed by tissue biopsy due to residual false-negative risk [58].

In metastatic breast cancer, ctDNA assays can non-invasively determine tumor genetics such as PIK3CA and ESR1 mutation status, which direct the use of targeted therapies (PI3K inhibitors like alpelisib for PIK3CA-mutant tumors, or novel endocrine therapies for ESR1-mutant cases). The concordance between LB and tissue for these mutations is high. For instance, one study comparing comprehensive genotyping found about 77% overall agreement between tissue DNA and ctDNA for PIK3CA mutations, rising to 95% concordance in patients with a ctDNA tumor fraction ≥ 2% [58]. This indicates that when sufficient tumor DNA is present in plasma, LB can virtually mirror tissue results. Importantly, some mutations are detected in blood that were missed in archived tissue, especially if the tissue test was limited (such as hotspot panel): up to 20–30% more PIK3CA-mutant cases could be identified by broad ctDNA sequencing relative to certain tissue assays. Recognizing these advantages, recent guidelines have incorporated LB in breast cancer management. The 2022 ASCO guidelines recommend plasma ctDNA testing for PIK3CA in hormone receptor-positive metastatic breast cancer, given its clinical utility and rapid turnaround [58]. In breast cancer, LB offers a gentler and more timely means to guide personalized therapy and may potentially detect relapse or resistance months earlier than standard imaging. Economic evaluations specific to breast LB are still emerging, but the qualitative benefits (fewer biopsies, quicker drug access) are clear, and these can translate into cost savings by avoiding unnecessary treatments.

Colorectal cancer (CRC) has likewise seen significant integration of liquid biopsy, particularly in the metastatic setting. A prime application is RAS (KRAS/NRAS) mutation testing in plasma to guide anti-EGFR therapy. Traditionally, RAS status is determined on tissue at diagnosis. However, tumor profiles can evolve under therapy and LB enables up-to-date genotyping without repeat tissue biopsies, which is crucial for selecting patients for EGFR inhibitors or rechallenge strategies. Studies have shown a high concordance between plasma ctDNA and tumor tissue for RAS mutations in metastatic CRC. In a large prospective trial (RASANC), the overall accuracy of blood-based RAS detection was 85% compared to tissue, and importantly rose to 95% in patients with sufficient ctDNA shedding (for instance those with active liver metastases) [102]. These data validate that, only under the right conditions, LB can reliably substitute for tissue in mutation testing. Moreover, LB often identifies resistance mutations earlier than conventional methods. Emergent RAS mutations or EGFR ectodomain mutations might be detected in blood months before radiographic progression in CRC patients on anti-EGFR antibodies [98]. However, the cost-effectiveness of routine ctDNA-guided therapy in early CRC is still under evaluation. A modeling analysis in stage II colon cancer found that using ctDNA to decide on adjuvant chemotherapy improved patient selection, but at the current test costs of about €67,000 per QALY (quality-adjusted life year) [103]. The model suggested that the ctDNA-guided strategy would become cost-effective if the assay cost fell below €1500 or if its predictive power increased [103]. In summary, LB in colorectal cancer allows for dynamic management, tracking tumor genomics in real time, and guiding therapy adjustments, with a high degree of fidelity to tissue analysis and an emerging framework for cost-effective use as technology matures.

Across NSCLC, breast, and colorectal cancers, liquid biopsy represents a transformative complement to traditional tissue biopsy. It provides a “liquid mirror” into tumor biology, offering rapid, repeatable access to molecular information that can inform targeted treatments, prognostication, and surveillance. The quantitative performance of LB has improved, approaching that of tissue: for many clinically actionable mutations, sensitivities in advanced disease range from about 70 to 85% with specificities of about 95–100%, and overall concordance rates between blood and tissue tests often exceed 85–90% [58,102]. While LB might not replace tissue histopathology and may miss low-shedding early tumors, its advantages in terms of accessibility, patient comfort, and turnaround time are compelling. The use of multiple analytes (ctDNA for mutations, CTCs for cellular analysis, EVs for nucleic acids/proteins, etc.) can broaden the scope of tumor characterization beyond what a single tissue sample provides. Early evidence also points to downstream economic benefits, such as reduced procedure costs and more efficient therapy allocation, when LB is incorporated judiciously [100,101]. In essence, LB versus tissue biopsy might not be necessary as an “either-or” proposition; rather, it might be a synergistic strategy: tissue biopsy establishes diagnosis and baseline tumor features, whereas LB contributes ongoing molecular snapshots throughout the disease course. By leveraging both approaches, clinicians can achieve a more complete and timely picture of the cancer, ultimately improving personalized treatment and patient outcomes [58].

## 3. Challenges in Liquid Biopsy

One of the major issues in LB is the difficulty of detecting the analytes in early-stage cancer and also the difficulty of evaluating the tumor molecular fraction. This factor is important to evaluate the yield of ctTNA, which is strictly dependent on the tumor burden and shedding, and varies according to the type of tumor, its location, and its stage, affecting the analytical sensitivity and specificity. It has been observed that DNA methylation can be used to estimate the tumor molecular fraction in DNA samples comparing the methylation profile between normal and cancer tissue [104]. In a study from 2022, Zhou et al. developed a method for early tumor diagnosis, analyzing and comparing the cfDNA methylation of healthy and cancer patients and identifying specific methylation sites in different kinds of cancer [105]. Recently, Gentzler et al. described a dynamic ctDNA biomarker that, using an algorithm, may measure changes in ctDNA quantitation and subsequently the alteration of the tumor molecular fraction [106]. The detection of low abundance ctDNA, ctRNA, and CTCs remains a significant challenge, as normal biomolecules can provide a high background due to the detection of subclones or the presence of clonal hematopoiesis (i.e., due to older age) [104,107].

Another crucial issue to discuss regards the standardization and validation of the methods used. Indeed, few LB tests are standardized, validated, and approved by the FDA or EMA, including the above-mentioned CellSearch^®^, cobas EGFR Test v2 [108], Guardant360 CDx [109], and FoundationOne liquid CDx [110]. The low number of standardized and validated protocols at both the pre-analytical and analytical level could lead to variability in experiments between different laboratories. For instance, there is sometimes the need to collect a large amount of plasma to extract enough ctDNA to perform the analysis, and there is currently no specific protocol that takes into account the type of tumor being studied and its intrinsic characteristics [111]. To overcome this kind of limitation, institutes from Europe to America, such as the European Liquid Biopsy Society (ELBS) and the Blood Profiling Atlas in Cancer (BloodPAC), are proposing guidelines on pre-analytical conditions, specifically for ctDNA analysis, addressing issues from quality control to DNA extraction kits and quantification [112,113].

Metastatic castration-resistant prostate cancer (mCRPC) and brain tumors are among the most challenging kinds of diseases to evaluate in LB, due to the capsular anatomy of the prostate and the blood brain barrier, which make ctDNA spreading challenging [114,115]. The limitations in this type of tumor arise from tumor shedding, which is influenced by tumor localization, altering the ctDNA kinetics and thus the molecular evaluability and the tumor fraction [116].

This concept leads to the study of cfDNA fragmentation, known as “fragmentomic”, which is a new field of study based on the examination of ctDNA fragments that are kept in the circulation by nucleosome protection and that represent the chromatin asset of the “native cell” in the tissue as well as genetic and epigenetic changes [117].

## 4. Future Prospects

### 4.1. Artificial Intelligence (AI) and Liquid Biopsy

Nowadays, the importance of AI is growing in several different areas, revolutionizing the way in which surgery, imaging, cancer diagnosis, and medicine are practiced as a whole [118]. It has been seen that machine learning (ML) algorithms can improve the identification of clinically relevant patterns in the biofluids and the diagnostic accuracy in clinical settings, integrating and synthetizing multidimensional omic data in order to classify the disease stage in different kinds of cancers, such as lung cancer [119], colorectal cancer [120], and meningioma [121], but also to identify the risk of developing prostate cancer based on gene expression from prostate tissue [118]. In essence, AI and ML may be trained to identify particular cancer patterns using datasets where the result is known, and then the result can be forecast by linking these features to novel and unidentified cancer data, thereby generating a likelihood score [122]. In this scenario, it is important to mention the CancerSEEK, a blood test that includes information on cfDNA/ctDNA mutations and protein biomarker concentrations. Using this test, it is possible to identify the mutation in cfDNA and the level of protein biomarkers for eight different types of cancer with a specificity higher than 99% and a sensitivity of about 70% [122]. Nowadays, AI finds its major application in diagnostic models for early-stage diagnosis. Recently, Li et al. trained an AI system called MethylBERT using over 100,000 cancer methylation data, in order to identify methylation markers in cfDNA samples from patients with epithelial ovarian cancer [123]. In the PROSTest study, Modlin et al. identified possible mRNA biomarkers in the whole blood of prostate cancer patients, using ML to identify significant features for distinguishing prostate cancer patients, outperforming the standard PSA test [124]. Recently, Karimzadeh et al. analyzed orphan circulating non-coding RNAs from the serum of over 1000 patients with NSCLC at different stages, demonstrating that a multi-task generative model, called Orion, is capable of sensitivity and specificity near 90% for early cancer detection and the classification of tumor subtypes, while also eliminating sources of noise unrelated to the disease status [125].

### 4.2. Multyanalyte and Multimodal Approaches

As previously mentioned, LB approaches often provide a low abundance of biomarkers, and the combination of different types (CTCs, ctDNA, EVs, ctRNA and TEPs) could offer a powerful method to provide a more comprehensive view of the tumor landscape and its characteristics, thereby improving prognosis, diagnosis, and monitoring, and helping to identify mechanisms of resistance developed after therapy [126,127].

LB, combined with other approaches, could be crucial for improving personalized therapy and for enhancing the detection of analytes. Further, this kind of approach can provide a more comprehensive understanding of tumor characteristics, overcoming the limitations of using a single approach [128,129].

In a bid to improve LB data and increase sensitivity compared to single-analyte approaches, Hofmann and colleagues recently combined CTCs characterization, mRNA expression analysis, and ctDNA analysis in patient samples with metastatic castration-resistant prostate cancer, detecting 89% of tumor-related information in the cohort analyzed. Despite the improved sensibility, a multi-analyte approach is influenced by the low abundance of biomarkers, such as CTCs and ctDNA [126].

In 2021, Keup et al. conducted a statistical analysis using blood samples from patients with metastatic breast cancer. They combined CTCs mRNA and gDNA, EVs mRNA and cfDNA, and highlighted how the analytes complement one another and provide crucial information on tumor heterogeneity [130]. Recently, Zhang and colleagues created a multi-omics method known as COMOS, which they evaluated using cfDNA taken from patients with diffuse large B-cell lymphoma. The work aims to obtain a fragmentomics landscape that includes nucleosome breakpoint characteristics (BSN), CpG islands (BSC), DNase clusters (BSD) and enhancers (BSE), methylation regions status (DMRs), and copy number alteration of cfDNA (CAN), in order to achieve high accuracy in early diagnosis and treatment response [131].

## 5. Discussion

The advantage of LB is that it provides a rapid, non-invasive, and cost-effective cancer assessment method, making it possible to perform the same analysis throughout the disease follow-up in order to monitor the disease status after treatment in the future. Though there are still obstacles to overcome, continued technological and methodological developments offer hope for broad clinical application. Furthermore, it is necessary to take into account the cost/efficacy ratio of using LB alone or in combination with other techniques. Recently, Malapelle et al. underscored the importance of evaluating both tissue and LB as complementary tests to provide a complete report capable of identifying two or more targetable markers [132]. With proper treatment monitoring, timely intervention, early detection, and accurate screening, LB has the potential not only for cancer diagnosis but also to become an essential component of customized cancer care as the field develops. In the future, the application of AI also in LB analyses could play a fundamental role in tailoring treatment for cancer patients, by making it possible to analyze the entire molecular profile and provide clinicians with a personalized strategy for patients [133]. Despite the potential role of AI, drawbacks need to be considered related to interpretability, the validation of AI models, and the ethical handling of patient data [134].

## 6. Conclusions

In conclusion, LB represents a transformative step toward precision oncology, offering a minimally invasive, repeatable, and comprehensive approach to tumor profiling. While analytes such as CTCs, cfDNA, and EVs are already being integrated into clinical workflows, their broader adoption depends on overcoming key challenges: early-stage sensitivity, method standardization, and biological complexity. Artificial intelligence and multianalyte strategies are emerging as powerful tools for enhancing diagnostic accuracy and interpretation. Thus, future research should focus on harmonizing pre-analytical protocols, validating biomarkers in large-scale studies, and ensuring equitable clinical implementation. When integrated with tissue analysis, LB may ultimately become a cornerstone of personalized cancer care.

## Figures and Tables

**Figure 1 ijms-26-05013-f001:**
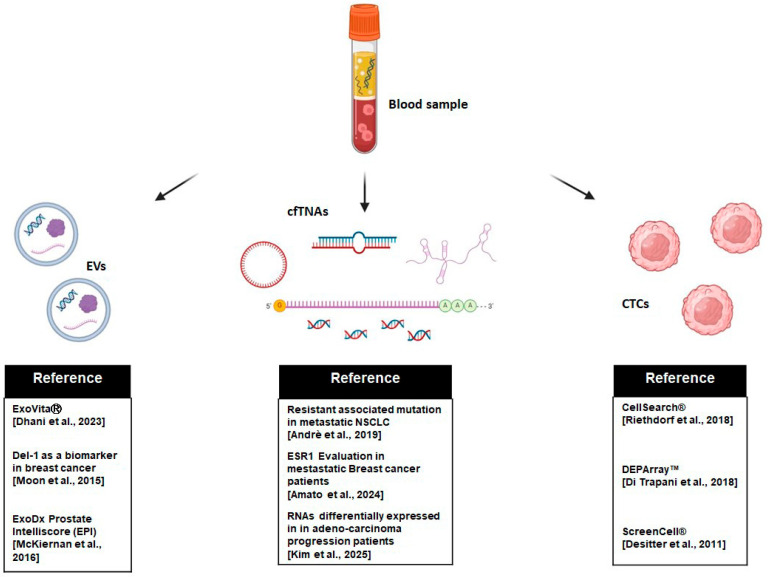
Infographic illustrating the analysis of three blood-derived biomarkers: extracellular vesicles (EVs) [86,87,88,89], circulating free tumor nucleic acids (cfTNAs) [56,58,76], and circulating tumor cells (CTCs) [25,31,35]. The figure highlights the respective detection methods and references from recent studies (created in BioRender.com).

**Table 1 ijms-26-05013-t001:** Summary of ESMO recommendations for the different kinds of genes already used in clinical practice.

Gene	Type of Mutation	Cancer Type	ctDNA Analysis Recommendation
*EGFR*	All pathogenic mutations	Non-Small Cell Lung Cancer	Recommended for cancer patients who have not responded to treatment and for individuals who have established resistance mutations to TKIs where tissue is unavailable or cannot be evaluated
*ALK*	Amplifications and fusions	Non-Small Cell Lung Cancer	Recommended only when tissue is unavailable or cannot be evaluated
*MET*			
*ROS1*			
*NTRK1-2-3*	Amplification, instability and fusions	Non-Small Cell Lung Cancer	Recommended only when tissue biopsy is not feasible or where prompt therapeutic decision-making is required
		Breast Cancer	
		Gastric Cancer	
		Pancreatic Cancer Hepatocellular Carcinoma	
		Cholangiocarcinoma	
		Urothelial Cancer	
		Soft Tissue Sarcoma	
		Thyroid Cancer	
*ERBB2*	Amplification, instability and fusions	Non-Small Cell Lung Cancer	Recommended only when tissue is unavailable or cannot be evaluated
		Breast Cancer	
		Gastric Cancer	
*PIK3CA*	All pathogenic mutations	Breast Cancer	Recommended for treatment monitoring in patients with progressing cancer
*BRCA1-2*	All pathogenic mutations	Breast Cancer	Recommended only when tissue is unavailable or cannot be evaluated and for treatment monitoring
		Ovarian Cancer	
		Prostate Cancer	
*ESR1*	All pathogenic mutations	Breast Cancer	Recommended for treatment monitoring
*MSI*	Instability	Breast Cancer	Recommended only when tissue is unavailable or cannot be evaluated
		Gastric Cancer	
		Pancreatic Cancer	
		Hepatocellular Carcinoma	
		Cholangiocarcinoma	
		Metastatic Colorectal Cancer	
		Ovarian Cancer	
		Endometrial Cancer	
		Prostate Cancer	
*IDH1*	All pathogenic mutations and fusions	Cholangiocarcinoma	Recommended only when tissue biopsy is not feasible or where prompt therapeutic decision-making is required
*FGFR2*			
*EGFR-ECD*	S492, G465, S464, V441	Metastatic Colorectal Cancer	Recommended for pre-treated patients with EGFR mutations
*KRAS/NRAS*	G12C	Non-Small Cell Lung Cancer	Recommended for patients who are naive to chemotherapy when a tissue biopsy is not available or when prompt treatment decision-making is required
	Exon 2, 3, 4	Metastatic Colorectal Cancer	
*BRAF*	V600E	Non-Small Cell Lung Cancer	Recommended for patients who are naive to chemotherapy when a tissue biopsy is not available or when prompt treatment decision-making is required
		Metastatic Colorectal Cancer	
		Thyroid Cancer	
*ATM*	Pathogenic mutations and deletions	Prostate Cancer	Recommended only when tissue is unavailable or cannot be evaluated
*PTEN*			
*PALB2*			
*FGFR*	All pathogenic mutations	Urothelial Cancer	Recommended only when tissue is unavailable or cannot be evaluated
*FGFR3*			
*RET*	Amplifications and fusions	Non-Small Cell Lung Cancer	Recommended only when tissue is unavailable or cannot be evaluated
		Thyroid Cancer	

**Table 2 ijms-26-05013-t002:** Liquid Biopsy methods: benefits, limitation and cost-effectiveness. Comparative overview of major methods used in liquid biopsy, including their biological source, clinical applications, technical benefits, and limitations, commonly used analytes, and an approximate assessment of cost-effectiveness based on current clinical use and scalability. Cost-effectiveness reflects a qualitative balance between clinical impact, technical complexity, and implementation cost.

Techniques	Analyte	Main Sources	Applications	Benefits	Limitations	Cost-Effectiveness
CellSearch^®^, DEPAr-ray™, ScreenCell^®^	CTCs	Blood	Prognosis, treatment monitoring, metastasis detection	Intact cells, allows for phenotypic/molecular analysis	Rare cells, complex enrichment, low yield	Moderate (high cost per result, limited use)
dPCR, NGS, Methylation analysis	cfDNA/ctDNA	Plasma/Serum	Mutation profiling for early diagnosis, MRD, resistance mutations	High sensitivity with NGS, widely used	Variable yield, interference from non-tumor DNA	High (approved tests available for certain cancer types, scalable)
RT-qPCR, RNA-Seq, ddPCR	cfRNA/ctRNA	Plasma/Urine	Biomarker discovery, early diagnosis, resistance mutations	Reflects active transcription, dynamic changes	RNA instability, technical noise	Moderate (emerging utility, limited protocols)
Ultracentrifugation, Immunoaffinity	EVs	Blood/Urine	Tumor cross-talk, early detection	High stability, rich content, present in many biofluids	Heterogeneous population, low specificity	Moderate (technically demanding, not standardized)
RNA-seq	TEPs	Blood	Early detection, immune evasion, prognosis	Easily accessible, reflects systemic tumor communication	Mechanisms not fully understood, RNA sequencing required	Low (experimental, no clinical translation yet)

## Data Availability

No new data were created or analyzed in this study. Data sharing is not applicable to this article.

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
