# Peer review of "Liquid Biopsy: The Challenges of a Revolutionary Approach in Oncology"

_ijms, 2025, doi:10.3390/ijms26115013_

Round 1

Reviewer 1 Report

Comments and Suggestions for Authors

Liquid biopsy, as a non-invasive diagnostic tool, has attracted extensive attention in the field of oncology. This review summarizes the current clinical application status of liquid biopsy in cancer and looks forward to its future, particularly emphasizing some methods used for the separation of liquid biopsy analytes. However, I think the author can further increase the comparison of the operation steps of different separation methods and try to analyze the advantages and disadvantages of different separation methods from the mechanism or principle as much as possible. Because the detection target substances in the separated samples are due to the LB technology.

Author Response

Liquid biopsy, as a non-invasive diagnostic tool, has attracted extensive attention in the field of oncology. This review summarizes the current clinical application status of liquid biopsy in cancer and looks forward to its future, particularly emphasizing some methods used for the separation of liquid biopsy analytes. However, I think the author can further increase the comparison of the operation steps of different separation methods and try to analyze the advantages and disadvantages of different separation methods from the mechanism or principle as much as possible. Because the detection target substances in the separated samples are due to the LB technology.

We thank the reviewer for the constructive comment. As suggested, we have compared the different separation method, including benefits and limits in a new table 2, discussing the methodology in the cost-effectiveness column. We also added a new paragraph to emphasize the comparison between liquid and tissue biopsy with sensitivity, specificity and cost-effectiveness data. Eventually, we added a conclusions paragraph to summarize a take-home message and highlight future perspectives more clearly.

Reviewer 2 Report

Comments and Suggestions for Authors

The article addresses a timely and relevant topic, but there are several areas that could be improved to enhance its scientific value and overall effectiveness:

  • Objectives and originality: while the topic of liquid biopsy is certainly important, the rationale behind the review is not clearly stated. I recommend better articulating the research question and clarifying the added value of this review compared to other recent narrative reviews on the same subject.
  • Structure and content: the manuscript mainly offers a descriptive overview of the available techniques, without adequately discussing their clinical advantages over tissue biopsy or providing quantitative data (e.g., sensitivity, specificity, cost-effectiveness). A more critical comparison supported by data would significantly strengthen the review.
  • Synthesis and readability: including a summary table comparing the different methods (e.g., clinical applications, benefits, limitations) would improve clarity and make the article more accessible and useful to readers.
  • Methodology: even in the context of a narrative review, it is important to report the article selection criteria and search strategy to ensure transparency and methodological rigor.
  • Conclusions: we suggest adding a concluding paragraph that clearly summarizes the key take-home messages and outlines future perspectives more effectively.

With targeted revisions, the manuscript could become a valuable and informative resource for readers interested in the evolving role of liquid biopsy in oncology.

Author Response

The article addresses a timely and relevant topic, but there are several areas that could be improved to enhance its scientific value and overall effectiveness:

  • Objectives and originality: while the topic of liquid biopsy is certainly important, the rationale behind the review is not clearly stated. I recommend better articulating the research question and clarifying the added value of this review compared to other recent narrative reviews on the same subject.
  • Structure and content: the manuscript mainly offers a descriptive overview of the available techniques, without adequately discussing their clinical advantages over tissue biopsy or providing quantitative data (e.g., sensitivity, specificity, cost-effectiveness). A more critical comparison supported by data would significantly strengthen the review.
  • Synthesis and readability: including a summary table comparing the different methods (e.g., clinical applications, benefits, limitations) would improve clarity and make the article more accessible and useful to readers.
  • Methodology: even in the context of a narrative review, it is important to report the article selection criteria and search strategy to ensure transparency and methodological rigor.
  • Conclusions: we suggest adding a concluding paragraph that clearly summarizes the key take-home messages and outlines future perspectives more effectively.

With targeted revisions, the manuscript could become a valuable and informative resource for readers interested in the evolving role of liquid biopsy in oncology.

We appreciate the reviewer’s insightful and detailed suggestions. In response:

Objectives and originality: We have substantially revised the manuscript to better articulate the objectives and originality of our review. We now clearly state the rationale and novelty in the Introduction.

Structure and content: We also included a new paragraph detailing the comparison between tissue and liquid biopsy supported by sensitivity, specificity and cost-effectiveness data.

Synthesis and readability: A new table 2 comparing the main liquid biopsy methods has been added, along with a more critical discussion including (benefits, disadvantages, cost-effectiveness) where available.

Methodology: The observation of the reviewer regarding methodology sound appropriate. However, we would like to underline the actual scope of the present review is to present the most important advancements regarding such a topic and not to perform a meta-analysis.

Conclusions: Finally, we added a conclusions section to highlight the key take-home messages and future perspectives more effectively.
